# Transcriptomic Analysis Identifies Molecular Response of the Tolerant Alfalfa (*Medicago sativa*) Cultivar Nongjing 1 to Saline-Alkali Stress

**DOI:** 10.3390/biology14040439

**Published:** 2025-04-18

**Authors:** Dongmei Zhang, Jinxia Li, Yiming Zhang, Yuanhao Zhang, Wenhui Wang, Zhaohui Li, Peng Zhu, Yongshun Huang, Long Han, Mingyu Wang, Zijian Zhang, Zhongbao Shen, Weibo Han, Linlin Mou, Xu Zhuang, Qiuying Pang, Jianli Wang, Lixin Li

**Affiliations:** 1Institute of Forage and Grassland Sciences, Heilongjiang Academy of Agricultural Sciences, Harbin 150086, China; zhangdongmei@haas.cn (D.Z.); shenzhongbao@haas.cn (Z.S.); hanweibo@haas.cn (W.H.); budaoweng@haas.cn (L.M.); yangyangcaocao@haas.cn (X.Z.); 2Key Laboratory of Saline-Alkali Vegetation Ecology Restoration, Ministry of Education, College of Life Sciences, Northeast Forestry University, Harbin 150040, China; lijinxia@nefu.edu.cn (J.L.); syzdxk@163.com (Y.Z.); zhangyuanhao1011@163.com (Y.Z.); wangwenhui0505@163.com (W.W.); lzhh010319@163.com (Z.L.); zhupeng@nefu.edu.cn (P.Z.); hallohys@163.com (Y.H.); hanlharry@163.com (L.H.); wmy19970825@163.com (M.W.); zhangzijian@163.com (Z.Z.); qiuying@nefu.edu.cn (Q.P.); 3College of Landscape Architecture, Northeast Forestry University, Harbin 150040, China

**Keywords:** *Medicago sativa*, saline-alkali adaptation, yield, flavonoid biosynthesis, transcriptional regulation

## Abstract

This study investigated the saline-alkali tolerance mechanisms in the tolerant alfalfa cultivar NQ-1 through transcriptomic analysis and metabolite profiling. Under saline-alkali stress, NQ-1 activates key metabolic pathways, including photosynthesis, α-linolenic acid metabolism, and flavonoid biosynthesis, which enhance its stress tolerance. Specific flavonoids and transcription factors potentially involved in regulating these pathways were identified. These findings advance our understanding of stress resistance mechanisms and could serve as a basis for the molecular design breeding of saline-alkali-tolerant alfalfa.

## 1. Introduction

Salinization and alkalization of soil significantly reduce the yields of crops, horticultural plants, and forages [1]. The Songnen Plain in Northeast China is a vital commodity grain production base and a major hub for animal husbandry. However, soil salinization in this region poses severe threats to both food security and the sustainable development of livestock farming. In this region, salinization and alkalization are primarily driven by carbonate accumulation. Extensive areas have degraded to moderately or severely saline-alkali soils, characterized by elevated pH, poor permeability, and low nutrient availability [2,3]. Breeding of saline-alkali-tolerant forage provides plant resources for ecological restoration and the sustainable utilization of soda saline-alkali lands [4,5,6,7].

The adaptation mechanisms of plants to saline-alkali stress are a complex biological process, involving physiological, biochemical, and molecular regulation at multiple levels [8,9,10,11]. The negative impacts of saline-alkali stress on plant biomass and yield are multifaceted. On the one hand, saline-alkali stress disrupts ion homeostasis, weakens root vitality [12,13], inhibits the photosystem, affects carbon fixation efficiency, and reduces stomatal conductance [14,15,16]. On the other hand, saline-alkali stress disrupts cellular pH homeostasis and membrane integrity. In order to adapt to stress environments, plants have evolved multiple mechanisms to alleviate the adverse effects of saline-alkali stress. The response mechanisms of plants to saline-alkali stress involve multi-level physiological and molecular regulations, including ion homeostasis, osmotic adjustment, antioxidant defense, and signal transduction. For example, plants maintain ion homeostasis through the SOS (Salt Overly Sensitive) pathway [17], NHXs (Na^+^/H^+^ antiporters), and HKTs (High-affinity K^+^ Transporters), which regulate the extrusion and compartmentalization of Na^+^ [18,19]. Osmotic adjustment is achieved by synthesizing osmoprotectants such as proline and soluble sugars [20,21]. The accumulation of reactive oxygen species (ROS) is alleviated by the antioxidant enzymes (e.g., SOD, CAT) and non-enzymatic antioxidants (e.g., ascorbate, glutathione) [22,23,24]. Moreover, calcium signaling [25,26], MAPK cascades [27,28], and plant hormones (e.g., JA, ABA) [29,30] play central roles in stress signal transduction, while epigenetic modifications (e.g., DNA methylation and histone modification) [31,32] further regulate the expression of stress-related genes. The coordinated action of these mechanisms provides a resistance strategy for plants to adapt to saline-alkali stress.

Flavonoids, a diverse group of polyphenolic compounds prevalent in plants, are categorized into several subclasses, including flavones, flavonols, flavanones, isoflavones, anthocyanins, chalcones, and flavan-3-ols [33]. These compounds exhibit antioxidant properties and regulate various metabolic processes. Under saline-alkali stress, flavonoids play a crucial role in enhancing plant stress tolerance. For instance, Isoliquiritigenin has antioxidant and cell-protective effects, mitigating oxidative damage to plant cells caused by reactive oxygen species (ROS) under saline-alkali stress [34]. Pelargonidin and cyanidin are anthocyanins that possess antioxidant properties and may help maintain normal photosynthesis in plants under saline-alkali stress by regulating plant pigment metabolism [35]. Formononetin is an isoflavone with antioxidant functions that may help maintain redox balance within plant cells under saline-alkali stress [36].

*Medicago sativa* is a highly valuable perennial deep-rooted forage legume, known for its widespread production, soil conservation capabilities, and ability to improve nitrogen-limited soils [37]. In this study, a high-quality cultivar, Nongjing 1 (NQ-1), suitable for cultivation in the northeast region was selected due to its saline-alkali resistance. Comparative transcriptome analysis of NQ-1 (saline-alkali vs. black soil) unveiled a total of 6013 differentially expressed genes (DEGs) involved in plant hormone signal transduction, photosynthesis, and flavonoid biosynthesis, etc., in response to saline-alkali stress. Our findings indicate that these pathways played crucial roles in NQ-1 stress resistance. This study deepened the understanding of the saline-alkali tolerance mechanisms of alfalfa.

## 2. Materials and Methods

### 2.1. Plant Materials and Treatments

Four alfalfa cultivars, namely, ‘Nongjing 1’ (NQ-1) and Control 1, Control 2, and Control 3, were provided by the Institute of Grassland, Heilongjiang Academy of Agricultural Sciences. The seeds were planted at two distinct experimental sites, Zhaodong (125°48′51.8832″ E, 46°15′20.7424″ N) and Harbin (126°50′51.5635″ E, 45°50′42.2469″ N), Heilongjiang Province, on 8 May 2023. The soil in Zhaodong is moderately sodic saline-alkali soil, with an average organic content of 13.47 g/kg and pH 8.5~9.0. The soil in Harbin is black soil, with an average organic content of 32.53 g/kg and pH 6.8~7.1. The regreening date in Zhaodong was 20~22 April 2024, while in Harbin, it was 5~10 April 2024. The regreening ratios are also documented in Figure 1A,B. The leaves were randomly sampled at the budding stage, on 8 June in Harbin and on 20 June in Zhaodong. The leaf tissues were flash-frozen in liquid nitrogen and stored at −80 °C.

### 2.2. Quantitative Real-Time PCR Validation

The gene-specific primers were designed using primer3Plus software (http://www.primer3Plus.com/, accessed on 26 November 2024), and the *MsActin* gene was used as the endogenous control. The RT-qPCR reactions were performed using the Agilent Mx3000P Real-Time PCR System (Agilent Technologies, Santa Clara, CA, USA). To test the specificity of the amplified products, the melting curve was generated and analyzed by increasing the temperature from 50 °C to 95 °C. Subsequently, the relative expression of genes was calculated using the 2^−ΔΔCT^ method. The gene specific primers are summarized in Appendix A. The total RNA used for quantitative real-time PCR validation was the same as that for RNA sequencing.

### 2.3. Metabolite Extraction

After the freeze-dried leaves were ground into a powder, 1.0 mL of 70% aqueous methanol was added to 100 mg of the powder, resuspended, and incubated overnight at 4 °C. Samples were centrifuged at 13,000 rpm, 4 °C for 15 min, and the supernatants were filtered with a 0.22 μm microfilter before LC-MS analysis.

### 2.4. Metabolite Detection

The leaf extracts were used for metabolic profiling by the previous method [38,39] with minor modification. The flow rate and column oven were set at 0.4 mL/min and 45 °C. The injection volume was 1 µL. Data acquisition was in full scan mode (70–1000 *m*/*z*) coupled with the IDA mode.

### 2.5. Physiological Measurements

Alfalfa leaves at the flowering stage (the first harvest) were sampled, and physiological and biochemical indices were analyzed as follows. Antioxidant enzyme activities (e.g., superoxide dismutase and peroxidase) and the proline (Proline) and malondialdehyde (MDA) contents were measured according to standard methods from the literature [40], with partial optimizations of the protocols. The chlorophyll content was determined using the acetone extraction method. Leaf tissue was ground and extracted with 80% acetone solution, and Chlorophylls a and b and the total content were calculated by measuring the absorbance at specific wavelengths (645 nm and 663 nm) using a spectrophotometer [41]. The lignin content was quantified via acetylation reaction combined with spectrophotometry, performed with the Acetylated Lignin Assay Kit (Acetylation Method) (G0708W, Suzhou Greiss Biotechnology Co., Ltd., Suzhou, China), and the content of jasmonic acid (JA) was quantified using a phytojasmonic acid enzyme-linked immunosorbent assay (ELISA) kit (KQ141769, Shanghai Scilink Biotechnology Co., Ltd., Shanghai, China). Three biological replicates per sample. Statistical significance was defined via one-way ANOVA, *p* < 0.05.

### 2.6. Total RNA Isolation, Library Construction, and RNA Sequencing

Total RNA was isolated and purified strictly following the standard protocol for the TRIzol reagent (Invitrogen, Carlsbad, CA, USA). RNA quality was analyzed by a NanoDrop 2000 spectrophotometer (Thermo Scientific, Waltham, MA, USA) for the absorbance rate and an Agilent 2100 Bioanalyzer (Agilent Technologies, Santa Clara, CA, USA) for the RNA Integrity Number (RIN). RNA Sequencing was conducted using the Illumina HiSeq X Ten (Illumina Inc., San Diego, CA, USA) platform according to a previous method with minor modification [42]. In total, 1 g of alfalfa leaves was used. For library construction, high-quality mRNA was enriched using poly (T) oligonucleotide probes and then fragmented using a chemical fragmentation buffer. Double-stranded cDNA was synthesized by reverse transcriptase. After end repair and poly(A) tailing, the cDNA libraries were sequenced by the MGIseq2000 platform (Metware Biotechnology Co., Ltd., Wuhan, China). Raw data were processed by Trimmomatic software (version 0.40) to remove low-quality reads and contaminants. Clean reads were aligned to the reference genome of *Medicago sativa* cultivar ‘ZhongMu No.1’ (http://alfalfagedb.liu-lab.com/alfalfa_download/), (accessed on 26 December 2024) using Hisat2. Gene expression levels were quantified as FPKM (Fragments Per Kilobase Million) values, calculated with the statistical model. The transcriptome original datasets have been deposited in the NCBI Sequence Read Archive (SRA) Database with reference number PRJNA1230803.

### 2.7. Enrichment Analysis of Differentially Expressed Genes

Differentially expressed genes (DEGs) were analyzed using the DESeq2 R package (version 1.20.0), with *p*-value < 0.05 and |log_2_FoldChange| ≥ 1. The expression patterns of DEGs in different samples were analyzed using hierarchical cluster analysis. The Gene Ontology (GO) (http://www.geneontology.org/, accessed on 14 November 2024) and KOBAS (KEGG orthology-based annotation system) (version 3.0) databases were used to analyze the GO enrichment and KEGG (Kyoto encyclopedia of genes and genomes) pathway enrichment of DEGs, respectively. KEGG pathway and GO enrichment analyses were based on *p*-value < 0.05.

## 3. Results

### 3.1. Nongjing 1 Is a High-Yielding, High-Quality, and Saline-Alkali-Tolerant Cultivar Suitable for Planting in Northeast China

Nongjing 1 (NQ-1), an alfalfa (*Medicago sativa*) cultivar adapted to Northeast China, was officially registered as a crop cultivar in Heilongjiang Province (Registration No. 2006007, 2006). A three-year (2003–2005) yield comparison trial was conducted between NQ-1 and the maternal cultivar LM803 (a crop cultivar registered in Heilongjiang Province, No. 133, 1993) at three experimental sites including Harbin (black soil, BS), Qinggang (sandy clay loam with weak alkaline, SCL), and Fuyu (weakly alkaline soil, WAS), Heilongjiang Province. The results demonstrated strong cold tolerance for both cultivars, with survival at winter temperatures as low as −35 °C and normal spring regreening. Under black soil conditions in Harbin, there was no significant difference in the regreening timing between NQ-1 and LM803 (Control 1) (Appendix A). Similarly, there was no significant difference in plant height or hay yield in the first harvest (Appendix A), indicating comparable growth performance under optimal soil conditions. However, under SCL and WAS soil conditions, NQ-1 exhibited a 2–3-day earlier regreening timing and a 0.1–1% higher regreening rate compared to LM803 (Appendix A). Additionally, the plant height and hay yield of NQ-1 were significantly higher than those of LM803 in the first harvest (Appendix A), indicating enhanced saline-alkali stress tolerance compared to the maternal parent LM803.

To evaluate their saline-alkali tolerance, NQ-1 and three control cultivars-LM803 (Control 1), LM801 (Control 2), and 218T (Control 3) -were planted in 2023 at two experimental sites in Heilongjiang Province: Harbin (black soil) and Zhaodong (moderate sodic saline-alkali soil, average pH 8.5). In black soil, the regreening ratios of LM803, LM801, and 218T were 3.2–5.5% lower than that of NQ-1 (95.2%). However, in the first harvest, there was no significant difference in the fresh weight between them (Figure 1B,C). In saline-alkali soil, the regreening ratios of NQ-1 and 218T maintained 95.30% and 94.82% of those in black soil, respectively. However, the fresh weight declined significantly to 69.34% and 70.73% of those in black soil in the first harvest. In contrast, LM801 and LM803 exhibited drastic reductions in the regreening ratio in saline-alkali soil (20.5% and 38.6%, respectively), with negligible yield (Figure 1A–C). These results classified NQ-1 and 218T as saline-alkali-tolerant cultivars, whereas LM801 and LM803 are saline-alkali-sensitive cultivars.

To elucidate the saline-alkali tolerance mechanisms of NQ-1, key physiological parameters were quantitatively analyzed. Under saline-alkali stress, the crude protein content of NQ-1 retained 78.98% of that in black soil (Figure 1D), indicating its resilience in productivity. The content of superoxide anion (O_2_^−^), a key reactive oxygen species (ROS) marker, increased by 27.69%, while the content of malondialdehyde (MDA), an indicator of oxidative membrane damage, increased by 65.9% (Figure 1E). These results demonstrate that saline-alkali stress disrupted ROS homeostasis and induced membrane lipid peroxidation. In response, the activities of two antioxidant enzymes-superoxide dismutase (SOD) and peroxidase (POD) -increased by 21.82% and 7.84%, respectively (Figure 1F), suggesting enhanced ROS scavenging capacity. The content of soluble sugar, a critical osmolyte for cellular osmotic balance, increased by 12.0%, whereas the levels of soluble proteins and proline remained unchanged (Figure 1G). Collectively, these findings demonstrate that NQ-1 has adopted multiple adaptive strategies for saline-alkali stress, balancing ROS scavenging, osmotic adjustment, and productivity maintenance.

### 3.2. Comparative Transcriptome Analysis of NQ-1 Leaves

To elucidate the molecular response mechanisms of NQ-1 to saline-alkali stress, we performed transcriptome profiling on leaf samples grown in black soil (BS, Harbin) and saline-alkali soil (SS, Zhaodong). RNA sequencing generated 107.22 million raw reads, with 104.53 million high-quality clean reads retained after stringent quality filtering. The clean reads exhibited a mapping rate exceeding 70% against the reference genome (Table 1), confirming the high-quality transcriptome assembly. These robust sequencing data provide a reliable foundation for downstream mechanistic exploration.

Principal component analysis (PCA) distinguished the two experimental groups, with a value of principal component 1 (PC1) of 47.2% and that of PC2 of 22% (Figure 2A). Comparative analysis identified 6013 differentially expressed genes (DEGs), with 3619 upregulated and 2394 downregulated in saline-alkali soil (SS) vs. black soil (BS) conditions (Figure 2B). Hierarchical clustering analysis of DEG expression patterns revealed distinct transcriptional profiles between the SS and BS groups (Figure 2C). The analysis of DEG classification indicated that in addition to the Signal transduction (T)-, Cell cycle (D)-, Translation (J-L)-, and Posttranslational modification (O)-related processes, Carbohydrate transport and mechanism (G), Secondary metabolite biosynthesis, transport, and catabolism (Q), and Intracellular trafficking, secretion, and vesicle transport (U) were also activated (Figure 2D), suggesting that photosynthesis, material transport, and secondary metabolism are important for the saline-sodic stress adaptation of alfalfa NQ-1.

The KEGG column chart revealed significant enrichment of DEGs in the Environmental Information Processing category, particularly in Plant hormone signal transduction and MAPK signaling pathway (Figure 3A, orange arrows). These findings were validated by the KEGG enrichment scatter plot (Appendix A). In the Metabolism category, 21.72% of DEGs were associated with the biosynthesis of secondary metabolites, including the isoflavonoid biosynthesis and flavone/flavonol biosynthesis pathways (Figure 3A, purple arrows). Notably, Carbon fixation in photosynthetic organisms and Starch and sucrose metabolism were the two most important pathways in the Metabolism category (Figure 3A, pink arrows). The KEGG enrichment chord plot (Figure 3B) displays the nine most significantly enriched pathways with the smallest *q*-values, each including ten DEGs with the highest |log_2_FC| values. For example, in the Biosynthesis of various plant secondary metabolites pathway, seven genes were upregulated and three downregulated (orange stars and bands), whereas the MAPK signaling pathway had four upregulated and six downregulated DEGs (red stars and bands). These results demonstrate that saline-alkali stress triggers the activation of interconnected pathways mentioned above to enhance the stress adaptation of NQ-1.

### 3.3. α-Linolenic Acid Metabolism Was Responsive to Saline-Alkali Stress

Jasmonic acid (JA) is an important phytohormone that plays a crucial role in plant growth and development and environmental stress response. Jasmonic acid is synthesized via the α-linolenic acid metabolism pathway (ko00592). JA is modified to form active ingredients including jasmonic acid isoleucine (JA-Ile) and methyl jasmonate (MeJA). JA regulates plant development, secondary metabolism, and stress response through the JA signaling pathway, which is mediated by a key transcription factor, *MYC2* (Figure 4A). Under saline-alkali stress, the expression levels of *fadA* and *ACX1*, which function in β-oxidization in peroxisomes, were significantly decreased (Figure 4A,B), and the JA content was also significantly downregulated (Figure 4C), suggesting that saline-alkali stress potentially inhibited the biosynthesis of JA and signal transduction.

### 3.4. Saline-Alkali Stress Affected the Photosynthesis of NQ-1

In nature, plants convert CO_2_ and H_2_O into organic compounds through photosynthesis in chloroplasts. Carbohydrates are mainly generated through the C3 and C4 pathways in the Carbon fixation in photosynthetic organisms (Ko00710) process, and the main products are sucrose and starch, etc. (Figure 5A). Saline-alkali stress affects plant photosynthesis through various ways such as the limitation of stomatal opening, degradation of photosynthetic pigments, inhibition of photosynthetic enzyme activity, damage to membrane systems, and accumulation of reactive oxygen species (ROS) [43,44,45]. Under saline-alkali stress, the transcription of *ALDO* (fructose-bisphosphate aldolase) and *PPC1* (phosphoenolpyruvate carboxylase1) was upregulated, and that of *FBP1* (fructose-1,6-bisphosphatase1) was downregulated (Figure 5A,C). The content of the photosynthetic pigment Chlorophyll b increased, while those of Chlorophyll a and carotenoids had no significant change, leading to a decrease in Chlorophyll a/b and carotenoids/Chlorophyll a+b (Figure 5D).

The Starch and sucrose metabolic pathway (ko00500) plays important roles in all biological processes such as plant growth and development, yield quality, and stress response, etc. (Figure 5B). In plants, starch is the most abundant carbohydrate reserve as the source of carbon and energy, and sucrose is an important factor controlling the transport tempo of nutrients, affecting crop growth, yield, and quality [46,47,48]. The metabolites in the pathway provide carbon sources, and some compounds act as signal molecules, e.g. trehalose (α-d-glucopyranosyl (1-1-)α-d-glucopyranoside) [49], which regulates carbon allocation and is involved in plant growth and development and the stress response [50]. Trehalose is synthesized from UDP-glucose catalyzed by trehalose-6-phosphate synthase (TPS) and trehalose-6-phosphate phosphatase (TPP) [51]. Under saline-alkali stress, the expression of *AMY* (α-amylase) and *TPP* (trehalose 6-phosphate phosphatase) is significantly increased (Figure 5B,C), suggesting a potential improvement in the synthesis capacity of trehalose.

### 3.5. Saline-Alkali Stress Did Not Inhibit Lignin Biosynthesis in NQ-1 Under Saline-Alkali Stress

The Phenylpropanoid biosynthesis (ko00940) pathway has multiple important functions in plants, and its products and intermediates play important roles in plant defense, signal transduction, etc. Some of them also have high medicinal activity. The pathway initiates from Phenylalanine to synthesize Cinnamoyl-CoA, which leads to the Flavonoid biosynthesis and Lignin biosynthesis pathways under the catalysis of different series of enzymes. PAL (phenylalanine ammonia-lyase) is the first rate-limiting enzyme catalyzing the reaction involving phenylalanine to produce Cinnamic acid, and its expression was significantly upregulated under saline-alkali stress (Figure 6A,B). CCRs (cinnamoyl-CoA reductases) catalyze the reduction of cinnamoyl-CoA to cinnamaldehyde, as well as the reduction of downstream products, e.g., p-Coumaraldehyde-CoA and Caffeoyal-CoA, which is a crucial step in lignin monomer synthesis. When upregulated, CCRs accelerate lignin biosynthesis, thicken plant cell walls, and enhance plant mechanical strength and stress resistance. Peroxidases play a role in the final stage of lignin synthesis, catalyzing the polymerization of *p*-Coumaryl alcohol and its downstream products to form various types of lignins (Figure 6A). When peroxidases are upregulated, the polymerization of lignins is promoted, which helps to thicken and reinforce the cell wall, enhancing plant mechanical strength and defense ability. Under saline-alkali stress, the expression of *CCR3* and *Peroxidase9* was significantly increased, while that of *Peroxidase12* was significantly decreased (Figure 6A,B). It is worth noting that the content of total lignins slightly decreased but had no significance (Figure 6C). Due to the inhibition of lignin biosynthesis by saline-alkali stress, the lack of significant changes in lignin in NQ-1 suggested its resistance to saline-alkali stress.

### 3.6. Flavonoid and Isoflavonoid Biosynthesis Pathways in NQ-1 Were Activated Under Saline-Alkali Stress

Flavonoids are widely and deeply involved in plant stress response. In the Flavonoid biosynthesis (ko00941) pathway, FLSs (flavonol synthases) catalyze the conversion of dihydroflavonol to flavonol (Figure 7A). Flavonols are strong antioxidants that protect cells from oxidative damage. When FLSs are upregulated, the flavonol contents increase and enhance plant resistance to biotic and abiotic stress, such as resisting pathogen invasion and UV damage. ANSs (anthocyanin synthases) catalyze the oxidation of colorless anthocyanins to produce anthocyanins (Figure 7A). When ANSs are upregulated, anthocyanins are synthesized in large quantities, causing plant organs to present rich colors and playing important roles in plant reproduction, defense, and response to environmental changes. CYP93Cs (cytochrome P450 family) are involved in the Isoflavonoid biosynthesis (ko00943) pathway (Figure 7B), participating in substrate conversion to affect the product types of isoflavones. When CYP93Cs are upregulated, the direction and rate of substrate conversion are changed and the specific isoflavones are increased, which may help plants cope with biotic and abiotic stress. Under saline-alkali stress, the expression levels of *FLS*, *ANS1*, *CYP75B1*-*5*, and *CYP93C2* were significantly increased (Figure 7A–C). Correspondingly, the content of total flavonoids also increased (Figure 7D), indicating that the Flavonoid and Isoflavonoid biosynthesis pathways were activated to protect NQ-1 and help it adapt to saline-alkali stress.

To verify the response of the biosynthesis of flavonoids and isoflavones to saline-alkali stress, we detected the contents of some flavonoids and isoflavones in leaves of NQ-1 and Control 3. As expected, the relative contents and trends of most of these metabolites in NQ-1 and Control 3 were similar (Figure 8A). Subsequently, we examined the contents of these metabolites in saline-alkali-tolerant and -sensitive cultivars grown in black soil. Surprisingly, there were significant differences between the two groups. For instance, the relative contents of Genistein, Cyanidin, and Pelargonidin in NQ-1 and C3 were significantly higher than those in C1 and C2, whereas Phlorizin, Isoliquiritigenin, Leucocyanidin, and Formononetin in NQ-1 and C3 were significantly lower than those in C1 and C2 (Figure 8B), revealing adaptive changes in flavonoid and isoflavonoid biosynthesis in saline-alkali-tolerant cultivars.

### 3.7. Identification and Analysis of Upstream Transcription Factors Potentially Functioning in Flavonoid and Isoflavonoid Biosynthesis in NQ-1 Under Saline-Alkali Stress

WGCNA analysis was performed to identify transcription factors that potentially regulate the flavonoid and isoflavonoid biosynthesis pathways (Figure 9A). The results showed that flavonoid and isoflavone biosynthetic enzyme genes were mainly enriched in the Brown, Blue, and Turquoise modules. In the Brown module, a total of 6 bHLHs and 12 MYBs transcription factors (TFs) were enriched, and their expression levels were significantly upregulated under saline-alkali stress (Figure 9B). There was a strong positive correlation (R > 0.9) between these TFs and the enzyme genes (Figure 9D). In addition, there were also correlations among the 6 bHLHs and 12 MYBs (Appendix A). Among them, the TFs *MsMYB_t2* and *MsbHLH_t3* showed the strongest correlation with multiple enzyme genes, suggesting that they may play a central role in regulating flavonoid and isoflavone biosynthesis in NQ-1 under saline-alkali stress. It is worth noting that the TFs *MsbHLH_b1* and *MsbHLH_b3* are only significantly correlated with *MsF6H_b1*, suggesting their transcriptional regulation specificity (Figure 9D). In addition, some TFs in the Tify, MADS-MIKC, AP2/ERF-AP2, NAC, and GARP families were also enriched in the Brown module. The strong correlation between the TFs *MsMYB*-*related_b1* and *MsbHLH_b6,* which had the highest expression levels among their gene families, suggested that they may also be involved in the regulation of flavonoid and isoflavone biosynthesis in NQ-1 under saline-alkali stress (Figure 9F). The expression levels of some key synthases and TFs in the Brown module were validated by RT-qPCR, and their trends were consistent with the results in transcriptome (Figure 9H).

In the Turquoise module, a total of 5 bHLHs and 25 MYBs were enriched, and the expression levels of these TFs were significantly downregulated under saline-alkali stress (Figure 9C). Correlation analysis showed a strong positive correlation (R > 0.9) between these TFs and enzyme genes in the same module, and correlations also existed among the transcription factors themselves (Appendix A). The TFs *MsMYB*-*related_t10* and *MsbHLH_t3* showed the strongest correlation with multiple enzyme genes, indicating their important roles in regulatory networks. In addition, *MsMYB*-*related_t1*, *MsMYB*-*related_t7*, and *MsMYB_t10* showed significant specificity in their correlation with specific enzyme genes (Figure 9E). Furthermore, some TFs in the NF-Y, C2C2, GRF, and b-ZIP families were also enriched in the Turquoise module. They showed strong correlations with *MsMYB*-*related_t10* and *MsbHLH_t3,* which had the highest expression levels among their gene families, suggesting that they may participate in the regulation of flavonoid and isoflavone biosynthesis in NQ-1 under saline-alkali stress (Figure 9G). The transcription levels of the key synthases and TFs in the Turquoise module were verified by RT-qPCR, and their trends were identical to those in the transcriptome analysis (Figure 9I). Furthermore, the expression levels of the above genes in saline-alkali-tolerant and -sensitive cultivars were also determined. The results indicate that *bHLH_b6*, *GARP*-*G2*-*like_b3*, *AP2/ERF*-*AP2_b3*, and *MYB*-*related_t10* exhibited different expression patterns in NQ-1 and C1 (Figure 9J), suggesting these TFs are related to the saline-alkali stress response in alfalfa.

## 4. Discussion

Photosynthesis produces carbohydrates and sugars for plant growth and survival in adversity. Chlorophyll is the core pigment of photosynthesis and is mainly responsible for absorbing red and blue light and converting it into chemical energy. Chlorophyll a is the main pigment directly involved in photosynthesis, while Chlorophyll b acts as an auxiliary pigment to help absorb a wider spectral range. The high excitation state of chlorophyll has a longer half-life and is prone to producing toxic reactive oxygen species (ROS). Carotenoids mitigate oxidative damage by scavenging ROS and absorbing complementary wavelengths (e.g., blue-violet light), transferring energy to chlorophyll. This pigment synergy optimizes photosynthetic efficiency and environmental adaptability [52].

The changes in the Chlorophyll contents and the rate of Chlorophyll a/b can reflect the physiological state and adaptability of plants. Under stress conditions, e.g., drought, salinity, and high temperature, chlorophyll-degrading enzymes are activated to accelerate the decomposition and destruction of chlorophyll molecules, which usually leads to a decrease in the contents of chlorophyll, resulting in a decrease in the efficiency of plant capture and transmission of light energy. Surprisingly, under saline-alkali stress, the content of Chlorophyll b increased while those of Chlorophyll a and carotenoids had no significant change, leading to a decrease in Chlorophyll a/b and carotenoids/chlorophyll in NQ-1 (Figure 5D). The stability of Chlorophyll a and carotenoids indicates that NQ-1 has a special mechanism to protect photosynthetic pigments from degradation, allowing it to have sufficient material and energy to resist the consumption caused by stress, thus maintaining most of the yield. If the protective mechanism of this photosynthetic pigment can be elucidated, it will greatly benefit the molecular breeding of saline-alkali-tolerant crops.

TPSs and TPPs are enzymes that are not only responsible for the biosynthesis of Trehalose but are also important for plant development and stress adaptation [53,54]. In quinoa, one CqTPP and the Class II CqTPSs are responsive to saline-alkali stress [50]. Under saline-alkali stress, in NQ-1, the expression of *AMY* (α-amylase) and *TPP* (trehalose 6-phosphate phosphatase) was significantly increased (Figure 5B,C), suggesting a potential improvement in the synthesis capacity of Trehalose, which may enhance saline-alkali stress tolerance.

Trehalose is a non-reducing disaccharide produced during starch and sucrose metabolism [51]. Trehalose regulates carbon allocation and plays an important role in plant growth, development, and stress response. The synthesis process of trehalose involves UDPG and glucose 6-phosphate (Glc6P) first being catalyzed by TPSs to produce trehalose-6-phosphate (T6P), which is then dephosphorylated by TPPs to produce trehalose [51]. UDPG and sucrose can be interconverted, mainly catalyzed by SUSs (sucrose synthases). T6P is not only a precursor of trehalose but also an important signaling molecule that regulates plant growth and development by dynamically regulating sucrose metabolism [55,56]. T6P mainly controls the activity of sucrose non-fermentative protein kinase 1 (SnRK1), and the two interact synergistically to maintain sugar homeostasis in plants [57,58]. TPSs and TPPs not only catalyze the biosynthesis of trehalose but also regulate plant stress resistance. For example, the expression level of *TPS* in wheat increases under saline-alkali stress, and the expression of *OsTPP1* in rice is also induced by saline-alkali stress. These changes in gene transcription levels can regulate plant stress tolerance [53,54]. Overexpression of *OsTPS1* in rice increases the content of glycosides and proline and promotes the upregulation of some stress response genes, thereby enhancing rice’s tolerance to cold, drought, and salinity [54,59]. Overexpression of the potato *TPS1* gene also enhances drought resistance [60]. Under saline-alkali stress, in NQ-1, the transcription of *SUS1/2*, *AMY*, and *TPP* was significantly increased (Figure 5B), suggesting that enhancement of trehalose biosynthesis may be one of the reasons for NQ-1’s saline-alkali tolerance.

Lignins enhance plant saline-alkali tolerance by reinforcing cell wall mechanical strength to resist osmotic stress and forming a physical barrier against salt infiltration. Their functional groups (e.g., phenolic hydroxyl) chelate metal ions, reducing salt toxicity [61]. Saline-alkali stress can induce glutathione S-transferase (GST), which promotes lignin accumulation by regulating SlCOMT2 activity and ROS scavenging [62]. Lignin biosynthesis-related factors may indirectly regulate the expression of ion transporters such as *HKT1* (Na^+^ efflux) and *SKOR1* (K^+^ influx), modulating ion selective transport, and thereby alleviate ion toxicity [63]. Notably, unlike the typical suppression of lignin biosynthesis under saline-alkali stress, NQ-1 maintained stable lignin levels, reflecting its strong adaptability.

Flavonoids directly scavenge ROS (H_2_O_2_, O_2_^−^, etc.) through active groups such as phenolic hydroxyl groups in their molecular structure while also activating antioxidant enzyme systems (e.g., SOD, POD), synergistically reducing oxidative stress. For example, in the Rosa salt-tolerant cultivar, flavonoid metabolites were significantly upregulated, and its ROS scavenging ability was positively correlated with salt tolerance. Flavonoids such as rutoside and quercetin help plants maintain metabolic homeostasis under saline-alkali stress by regulating energy metabolism and carbon-nitrogen allocation. For example, flavonoid metabolites in the Rosa salt-tolerant cultivar synergistically regulate the stability of cell membranes with lipids and phenolic acids [64]. Flavonoids enhance plant adaptability to saline-alkali stress by regulating the expression of antioxidant enzyme genes. For example, in *Medicago truncatula*, β-glucosidase 17 (MtBGLU17) hydrolyzes flavonoid glycosides to release active aglycones, enhancing the antioxidant capacity and thus improving salt tolerance [65] In addition, flavonoids in the root system reduce the transport of Na^+^ to the aerial part by chelating Na^+^ or regulating the expression of ion transporters (such as HKT1), maintaining the intracellular Na^+^/K^+^ balance. For example, MtBGLU17 in alfalfa regulates ion selective absorption through flavonoid glycosides, alleviating ion toxicity caused by saline-alkali stress [65].

Under saline-alkali stress, plants regulate the expression of flavonoid biosynthesis genes through transcription factors such as MYB, bHLH, etc. Tobacco (*Nicotiana tabacum*) *NtMYB4* is a transcription repressor whose expression is inhibited under saline-alkali stress, relieving the inhibition of the chalcone synthase gene (NtCHS1) and promoting flavonoid biosynthesis [66]. In most species, flavonoid biosynthesis is inhibited under saline-alkali stress. However, in some tolerant cultivars, flavonoid biosynthesis is activated and the total flavonoid content increases, such as the salt-tolerant Rosa cultivar mentioned above. NQ-1 is also a saline-alkali-tolerant cultivar of this type, with enhanced flavonoid biosynthesis and total flavonoid content. In addition, we have detected specific flavonoid types that significantly increase under saline-alkali stress, providing information for molecular breeding aimed at generating saline-alkali-tolerant alfalfa by increasing the contents of resistant flavonoids.

## 5. Conclusions

In this study, we focused on the saline-alkali-tolerant molecular response including important pathways, key genes, and metabolites in Nongjing 1, a stress-tolerant alfalfa cultivar. Comparative transcriptome analysis of NQ-1 grown in moderate saline-alkali soil vs. black soil identified a total of 6013 differentially expressed genes (DEGs), which were mainly enriched in α-Linolenic acid metabolism, Carbon fixation in photosynthetic organisms, Starch and sucrose metabolism, Phenylpropanoid biosynthesis, and Flavonoid biosynthesis pathways. Moreover, the upstream transcription factors that potentially participate in the regulation of Flavonoid biosynthesis were identified. The expression levels of the key genes and the accumulation levels of the metabolites were detected, suggesting that these synthases, TFs, and metabolites were crucial for the stress resistance of NQ-1. Our findings provided valuable information for understanding the stress resistant mechanism of alfalfa, which can serve as potential target genes for molecular design breeding strategies for saline-alkali-tolerant alfalfa.

## Figures and Tables

**Figure 1 biology-14-00439-f001:**
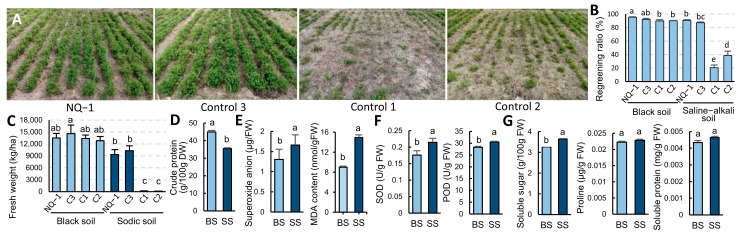
Growth performance and physiological indices of NQ-1 and the control cultivars. (**A**) Phenotypic characteristics of NQ-1 and the control cultivars during the regreening stage in soda saline-alkali land in Zhaodong, Heilongjiang Province. (**B**) Statistics of the regreening ratio of the cultivars. (**C**) Statistics of the fresh weight in the first harvest. (**D**–**G**) Physiological index results. The leaves were sampled at the initial flowering stage and used for determination of crude protein (**D**), superoxide anion and MDA content (**E**), antioxidant enzyme activities (**F**), soluble sugar content, proline content, and soluble protein (**G**). Data are means ± SE, *n* ≥ 3. The different letters indicate a significant difference, *p* < 0.05.

**Figure 2 biology-14-00439-f002:**
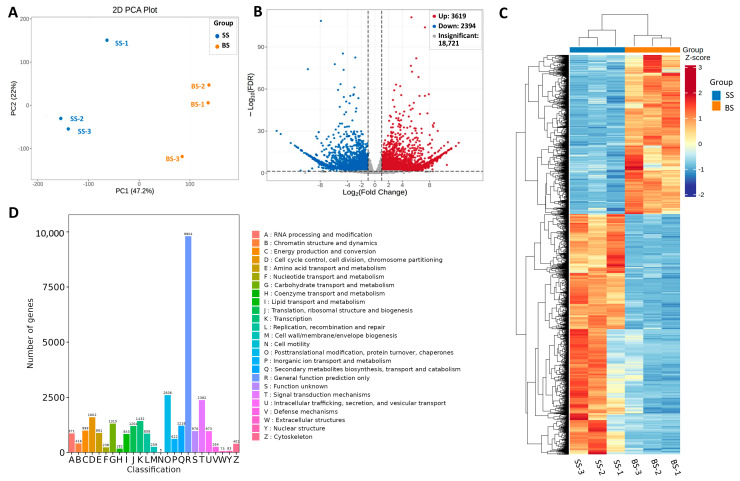
General overview of differentially expressed genes (DEGs). (**A**) PCA diagram. Blue dots, SS group; orange dots, BS group. X- and Y-axes, the Principal Component 1 (PC1) and Principal Component 2 (PC2). Scores of PC1 and PC2 indicate cohesion within each group and separation between the SS and BS groups, respectively. (**B**) Volcano map of the genes of SS vs. BS. Red dots, upregulated genes; blue dots, downregulated genes; gray dots, genes with no significance. (**C**) Hierarchical clustering thermogram of DEGs for SS vs. BS. (**D**) Quantitative statistics of Eukaryotic Orthologous Groups (KOG) functional classification of DEGs.

**Figure 3 biology-14-00439-f003:**
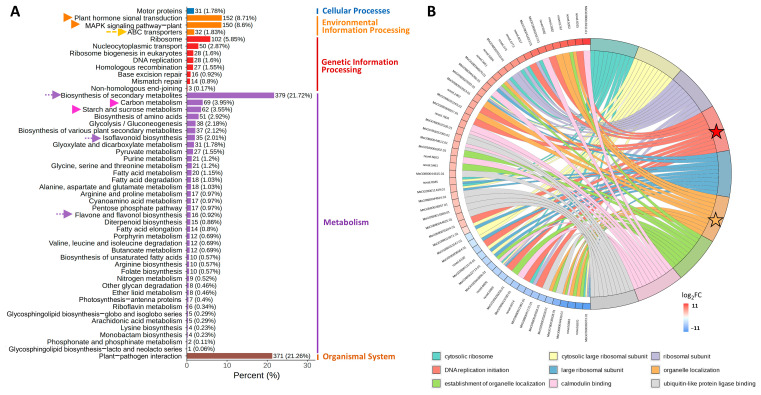
KEGG enrichment analysis for SS vs. BS. (**A**) Column chart analysis. (**B**) Chord plot analysis. The left half contains ten DEGs with the highest |log_2_FC| values in each pathway, while the right half contains the nine pathways with the smallest *q*-values.

**Figure 4 biology-14-00439-f004:**
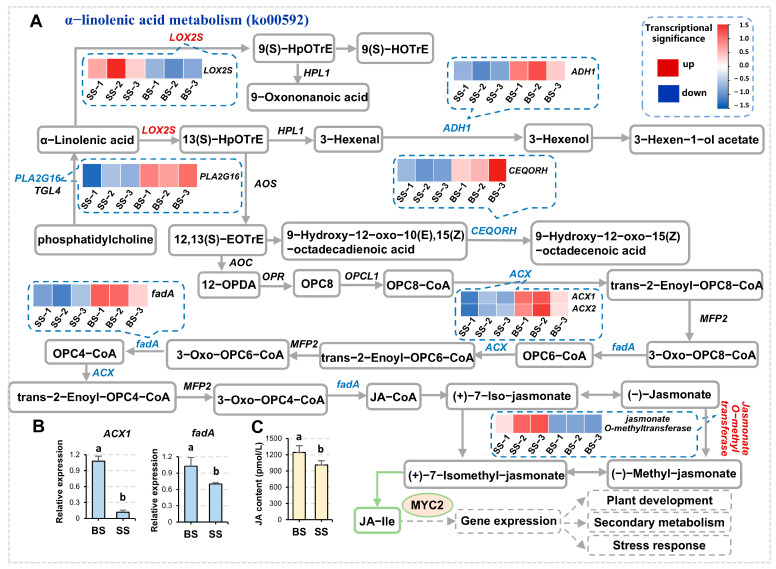
The α-linolenic acid metabolism pathway in response to saline-alkali stress (SS vs. BS, the same below). (**A**) Part of the α-linolenic acid metabolism pathway response to saline-alkali stress. The heatmap reflects the changes in gene expression levels of major enzymes quantified in a color scale. Gray frames, the metabolites; solid lines with arrows, directions of the processes; dashed lines with arrows, downstream pathways. Red names, upregulated gene families; blue names, downregulated gene families; green names, gene families including both upregulated and downregulated members (the same representation is used in the following figure legends, so these are omitted below). (**B**) Statistics of the expression levels of the indicated genes validated by RT-qPCR. *MsActin* was used as an endogenous control. Three independent experiments per sample, and three technical replicates per experiment were performed. BS, black soil; SS, saline-alkali soil (the same below). (**C**) Statistics of JA contents. Data are means ± SE of three biological replicates (*n* ≥ 3). The letters above the error bars indicate significance at *p* < 0.05.

**Figure 5 biology-14-00439-f005:**
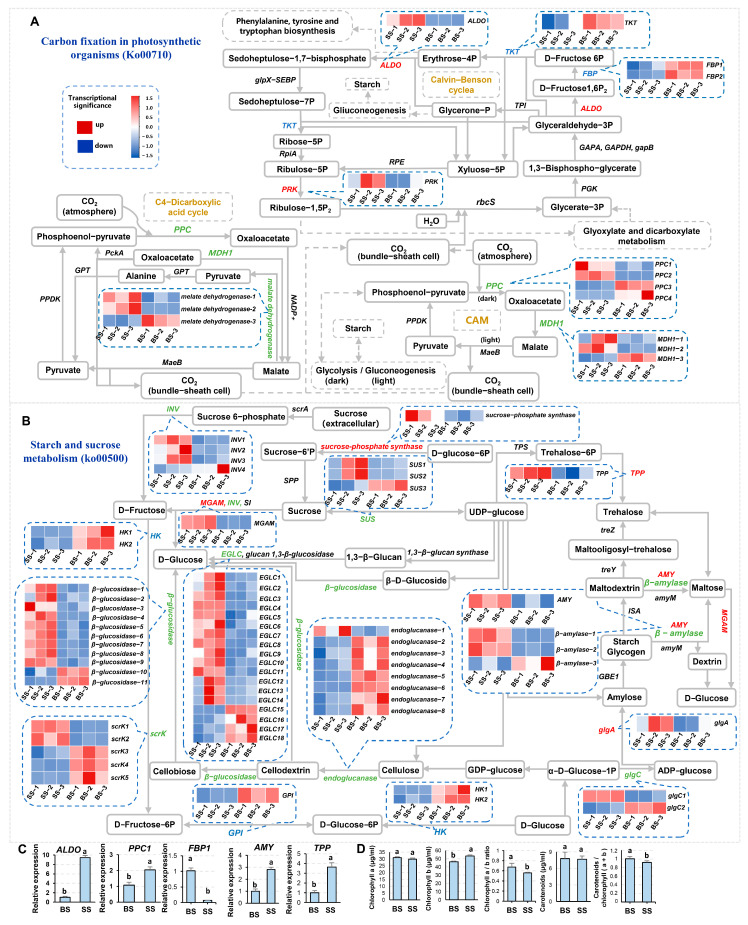
Carbon fixation in photosynthetic organisms and starch and sucrose metabolism pathways in response to saline-alkali stress. (**A**,**B**) Part of the indicated pathways of response to saline-alkali stress. (**C**) Statistics of the gene expression levels validated by RT-qPCR. *MsActin* was used as an endogenous control. Three independent experiments per sample, and three technical replicates per experiment were performed. (**D**) Statistics of chlorophyll and carotene contents. Data are means ± SE, *n* = 3. The letters above the error bars indicate significance at *p* < 0.05.

**Figure 6 biology-14-00439-f006:**
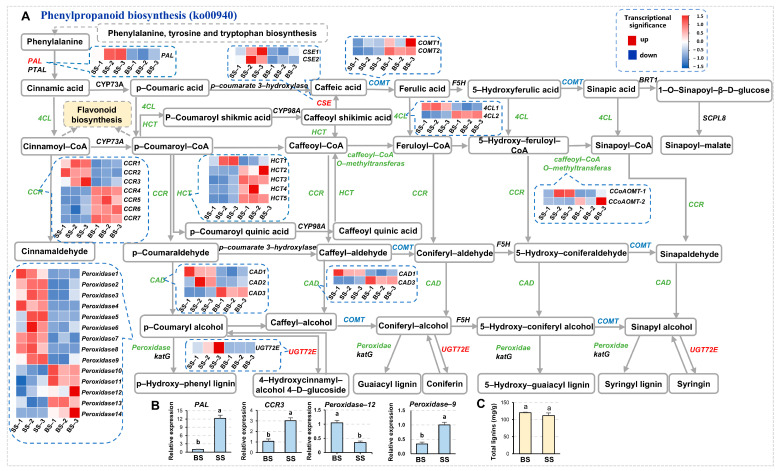
Phenylpropanoid biosynthesis pathway in response to saline-alkali stress. (**A**) Part of the Phenylpropanoid biosynthesis pathway response to saline-alkali stress. (**B**) Statistics of the gene expression levels validated by RT-qPCR. *MsActin* was used as an endogenous control. Three independent experiments per sample, and three technical replicates per experiment were performed. (**C**) Statistics of total lignin contents. Data are means ± SE, *n* ≥ 3. The letters above the error bars indicate significance at *p* < 0.05.

**Figure 7 biology-14-00439-f007:**
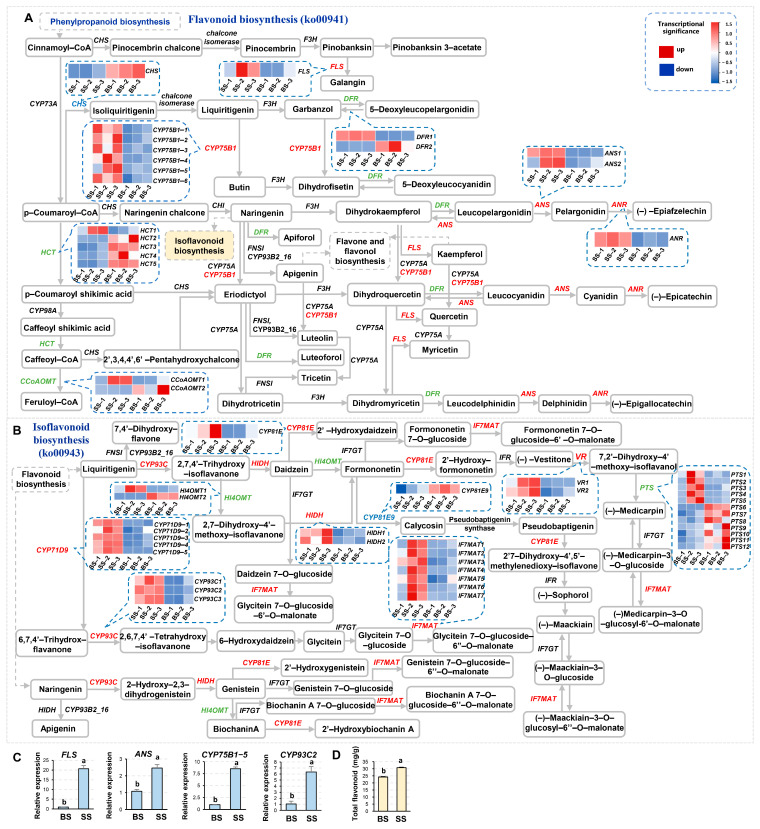
Flavonoid biosynthesis and Isoflavonoid biosynthesis pathways in response to saline-alkali stress. (**A**,**B**) Part of the indicated pathways responses to saline-alkali stress. (**C**) Statistics of the gene expression levels validated by RT-qPCR. *MsActin* was used as an endogenous control. Three independent experiments per sample, and three technical replicates per experiment were performed. (**D**) Statistics of total flavonoid contents. Data are means ± SE, *n* ≥ 3. The letters above the error bars indicate significance at *p* < 0.05.

**Figure 8 biology-14-00439-f008:**
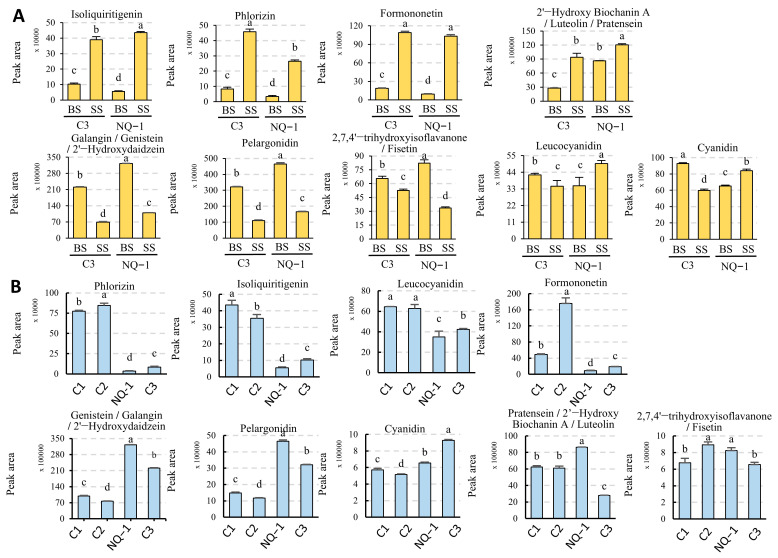
Detection of flavonoid compounds by LC-MS. (**A**) Comparison of flavonoid contents between NQ-1 and Control 3 under saline-alkali stress. (**B**) Comparison of flavonoid contents between NQ-1 and Controls 1-3 under black soil conditions. The peak areas were obtained from LC-MS detection in negative ion mode and normalized by internal standards. Data are means ± SE, *n* ≥ 3. The letters above the error bars indicate significance at *p* < 0.05. Information for the compounds: Phlorizin (*m/z* 435.1297), Isoliquiritigenin (*m/z* 255.0663), Leucocyanidin (*m/z* 305.0667), Formononetin (*m/z* 267.0663), Genistein/Galangin/2′-Hydroxydaidzein (*m/z* 269.0455), Pelargonidin (*m/z* 270.0534), Pratensein/2′-Hydroxy Biochanin A/Luteolin (*m/z* 299.0561), Cyanidin (*m/z* 286.0483), 2,7,4′-trihydroxyisoflavanone/Fisetin (*m/z* 271.0612).

**Figure 9 biology-14-00439-f009:**
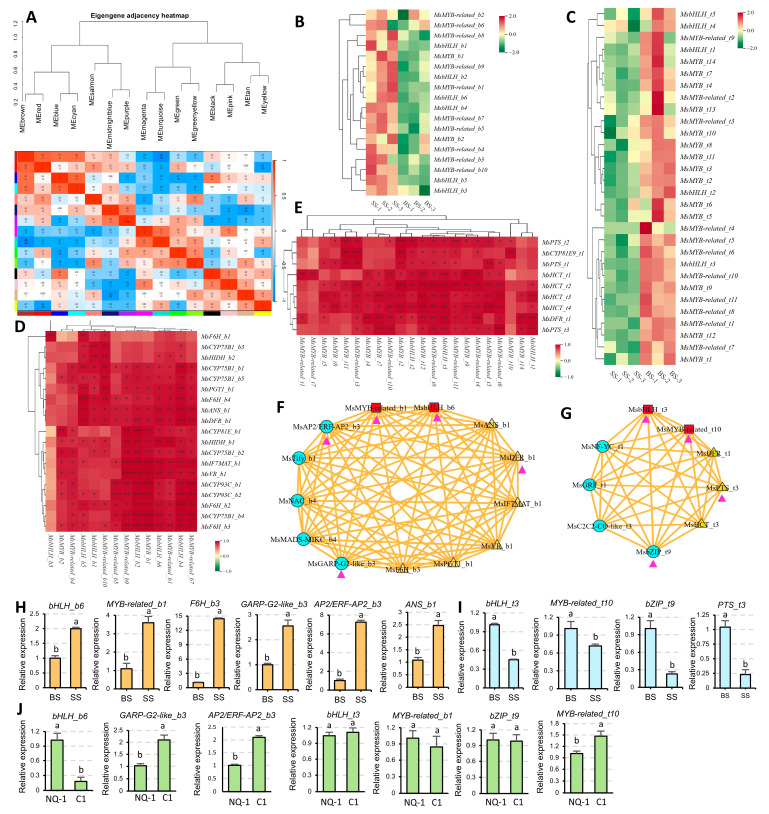
Identification and analysis of upstream transcription factors potentially regulating flavonoid and isoflavone biosynthesis. (**A**) Transcriptomic WGCNA analysis. (**B**,**C**) Heatmaps of MsbHLH and MsMYB genes in Brown (**B**) and Turquoise (**C**) modules. (**D**,**E**) Functional correlation analysis between MsbHLH, MsMYB, and synthases in the Brown (**D**) and Turquoise (**E**) modules. Correlation analysis was conducted using the Pearson method. *, *p* < 0.05; **, *p* < 0.01; ***, *p* < 0.001. (**F**,**G**) Functional correlation analysis of specific MsbHLH and MsMYB pairs (*MsbHLH_b6* and *MsMYB*-*related_b1*; *MsbHLH_t3* and *MsMYB*-*related_t10*) with target synthases in respective modules. (**H**,**I**) The expression levels of MsbHLHs, MsMYBs, and synthase genes in Brown (**H**) and Turquoise (**I**) modules were verified by RT-qPCR. (**J**) The expression levels of the genes in Brown and Turquoise modules in saline-alkali-tolerant and -sensitive cultivars were verified by RT-qPCR. *MsActin* was used as an endogenous control. Three independent experiments per sample, and three technical replicates per experiment were performed. The letters above the error bars indicate significance at *p* < 0.05.

**Table 1 biology-14-00439-t001:** Statistics of sequencing output and mapping ratio.

Sample	Raw Reads (Mb)	Clean Reads (Mb)	Clean Base (G)	Reads Mapped	Unique Mapped
BS	54.93	53.56	8.04	42.90 (80.07%)	40.49 (75.58%)
SS	52.29	50.97	7.32	39.83 (78.14%)	37.66 (73.89%)

## Data Availability

The data presented in this study are openly available in NCBI at https://www.ncbi.nlm.nih.gov/bioproject/PRJNA1230803 (accessed on 12 March 2024), reference number PRJNA1230803.

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
