# Peer review of "Transcriptomic Analysis Identifies Molecular Response of the Tolerant Alfalfa (Medicago sativa) Cultivar Nongjing 1 to Saline-Alkali Stress"

_biology, 2025, doi:10.3390/biology14040439_

Round 1
Reviewer 1 Report
Comments and Suggestions for Authors
The manuscript by Zhang et al. presented molecular responses of saline-alkali tolerant alfalfa variety to saline-alkali conditions. The authors performed multi-year field experiments and identified two tolerant varieties, NQ-1 and 218T, and two sensitive varieties, LM801 and LM8013. Phenotyping with various parameters was convincingly indicated that NQ-1 is a saline-alkali tolerant line and thus was chosen for further transcriptomics study. The authors compared the transcriptomes of saline-alkali soil grown and normal soil grown NQ-1 leaves and identified several pathways that could be responsible for its saline-alkali tolerance. These pathways include photosynthesis and secondary metabolisms. Notably, the authors identified that flavonoids accumulation and potential transcription factors modulating flavonoids biosynthesis pathway may contribute to the tolerance of NQ-1.
Overall, this study is significant in identifying saline-alkali tolerance alfalfa that could be readily used in agricultural production in saline-alkali affected areas. Furthermore, transcriptomic analysis provided data for better understanding of the molecular response of this tolerant line to saline-alkali conditions. However, it needs to address some concerns before its publication.
- The title used “comparative” leading to the assumption that this study compared the transcriptomes of the tolerant line vs. the sensitive line to identify differentially expressed genes and thus further identify the pathways and key components responsible for saline-alkali tolerance, which is a common way for such approaches. However, the present study only analyzed the transcriptomes of tolerant NQ-1 line with saline-alkali stress vs. without saline-alkali stress. Therefore, the title should be changed to something like “Transcriptomic analysis identifies molecular response of the tolerant alfalfa variety Nongjing 1 to saline-alkali stress”.
- Related to point #1, the identified DEGs (stress vs non-stress) and their associated pathways are not convincible as responsible for the tolerant phenotype, unless the authors also include the results from the sensitive line in figures such as Figure 4B, 4C, Figure 5C, 5D, Fig. 6 B-C, Fig. 7C-D. If this is not possible, the authors are suggested to tone down the conclusions. It is important to note that some sensitivity may be caused by over-response at the gene expression level, and thus increased expression of genes does not necessarily mean these genes are responsible for tolerance.
- It is good to see that Figure 8 included control lines to compare with NQ-1, and the conclusion that flavonoids are likely to contribute to the tolerant phenotype of NQ-1. If the authors want to draw the same conclusion for the TFs based on Figure 9, the Figure H-I should also include control lines. Otherwise, the conclusion needs to be toned down.
- Other minor points: in the Results, line 89, “for the first time” means what? The sentence “and plant height and hay weight of NQ-1 harvested for the first time also had no significance” needs to be revised; Line 166, “TOP2” means “top two”? subtitle 2.5, “SchemesSaline-alkali stress”?
Some language editing is needed as indicated in my review report.
Author Response
Please see the attachmen.

Reviewer 2 Report
Comments and Suggestions for Authors
In this study, the authors performed transcriptomic analysis to compare the gene expressions between saline-alkali tolerant alfalfa variety NQ-1 and saline-alkali sensitive varieties. They found that genes involved in photosynthesis and secondary metabolic pathways are highly expressed in the NQ-1 variety, which is supposed to contribute to the salt tolerance of NQ-1 variety. Furthermore, they identified transcription factors that regulate flavonoid biosynthesis in NQ-1. Overall, this study will help us understand how plants, especially alfalfa, adapt to salt stress in a molecular and metabolic ways.
Major comments:
- Gene and metabolite names, as well as abbreviations, must be annotated in full. Ensure consistency between figures and the main text.
- The current version of the manuscript requires thorough proofreading for grammar and spelling errors.
- Please clearly specify the plant tissues used in this study.
- The experimental methods (e.g., section 4.5) should be appropriately elaborated.
- Figure S2 and Figure S3 are not mentioned in the main text—please address this.
- For the key flavonoid compounds detected, introduce their reported functions and explain why these substances were the focus of the study.
- The Discussion section should be appropriately condensed, focusing on key candidate genes involved in flavonoid biosynthesis in response to saline-alkali stress.
English Language needs to be further improved.
Author Response
Please see the attachmen.
